# Mixed-Valent Trinuclear Co^III^-Co^II^-Co^III^ Complex with 1,3-Bis(5-chlorosalicylideneamino)-2-propanol

**DOI:** 10.3390/molecules27134211

**Published:** 2022-06-30

**Authors:** Masahiro Mikuriya, Yuko Naka, Ayumi Inaoka, Mika Okayama, Daisuke Yoshioka, Hiroshi Sakiyama, Makoto Handa, Motohiro Tsuboi

**Affiliations:** 1School of Biological and Environmental Sciences, Kwansei Gakuin University, Uegahara 1, Gakuen, Sanda 669-1330, Japan; poninma@gmail.com (Y.N.); ayumi.inaoka.570@gmail.com (A.I.); sym12286@gmail.com (M.O.); yoshi0431@gmail.com (D.Y.); tsuboimot@kwansei.ac.jp (M.T.); 2Department of Science, Faculty of Science, Yamagata University, 1-4-12 Kojirakawa, Yamagata 990-8560, Japan; saki@sci.kj.yamagata-u.ac.jp; 3Department of Chemistry, Graduate School of Natural Science and Technology, Shimane University, 1060 Nishikawatsu, Matsue 690-8504, Japan; handam@riko.shimane-u.ac.jp

**Keywords:** trinuclear complex, cobalt complex, mixed-valent complex, Schiff-base ligand

## Abstract

A mixed-valent trinuclear complex with 1,3-bis(5-chlorosalicylideneamino)-2-propanol (H_3_clsalpr) was synthesized, and the crystal structure was determined by the single-crystal X-ray diffraction method at 90 K. The molecule is a trinuclear Co^III^-Co^II^-Co^III^ complex with octahedral geometries, having a tetradentate chelate of the Schiff-base ligand, bridging acetate, monodentate acetate coordination to each terminal Co^3+^ ion and four bridging phenoxido-oxygen of two Schiff-base ligands, and two bridging acetate-oxygen atoms for the central Co^2+^ ion. The electronic spectral feature is consistent with the mixed valent Co^III^-Co^II^-Co^III^. Variable-temperature magnetic susceptibility data could be analyzed by consideration of the axial distortion of the central Co^2+^ ion with the parameters Δ = –254 cm^−1^, *λ* = –58 cm^−1^, *κ* = 0.93, *tip* = 0.00436 cm^3^ mol^−1^, *θ* = –0.469 K, *g*_z_ = 6.90, and *g*_x_ = 2.64, in accordance with a large anisotropy. The cyclic voltammogram showed an irreversible reduction wave at approximately −1.2 V·vs. Fc/Fc^+^, assignable to the reduction of the terminal Co^3+^ ions.

## 1. Introduction

Schiff-base ligands have been synthesized extensively because such organic compounds may be easily accessible for constructing various kinds of multidentate ligands and useful for reacting with main-group and transition metal ions, including lanthanides and actinides, to form a number of metal complexes, which are useful as model compounds in basic chemistry as well as in a wide range of applications [1,2,3,4,5,6,7,8,9,10]. Pentadentate Schiff-base ligands, 1,3-bis(salicylideneamino)-2-propanol (H_3_salpr), and its substituted derivatives have been developed as dinucleating ligands for constructing adjacent coordination sites, as shown in the case of 1,3-bis(5-chlorosalicylideneamino)-2-propanol in Figure 1a [11,12,13,14,15,16,17,18,19,20,21,22,23,24,25,26,27,28,29,30,31,32,33,34,35,36,37,38,39,40,41,42,43,44,45,46,47,48]. X-ray crystal structure analysis was performed for some free Schiff-base ligands [11,12,13,14,15,16]. In the crystals, the Schiff bases take a “bent” [11,12,13,14,15,34] or “folded” [16] structure with the salicylideneaminomethyl moieties being close to planar. When the Schiff-base ligands are coordinated to two metal atoms with two phenolic-oxygen, two imino-nitrogen, and one bridging alkoxido-oxygen donor atom with a pair of tridentate *O*, *N*, *O*-chelates, Mn^III^_2_, Fe^III^_2_, Co^III^_2_, Ni^II^_2_, and Cu^II^_2_ complexes were reported as this type of dinuclear species [17,18,19,20,21,22,23,24,25,26,27,28,29]. On the other hand, these Schiff-base ligands form mononuclear metal species with a tetradentate *O*, *N*, *N*, *O*-chelate as shown in the case for 1,3-bis(5-chlorosalicylideneamino)-2-propanol in Figure 1b, where the central alcohol group remains protonated and does not participate in coordination to the metal center. Such mononuclear Mn^III^, Ni^II^, Cu^II^, and Pd^II^ complexes were found in the literature [30,31,32,33,34,35,36], and a hydrolyzed product of mononuclear Co^III^ species was also derived from a reaction of a Schiff-base ligand with cobalt salt [37]. In the other case, the mononuclear species are further connected to form trinuclear Cu^II^_3_ [36], Zn^II^_3_ [38], Cd^II^_3_ [39], and Co^II^Co^III^_2_ [40]; tetranuclear Mn^II^_2_Mn^III^_2_ [42,43], Mn^III^_4_ [44], Co^II^_4_ [39,45], Co^II^_2_Co^III^_2_ [46], Ni^II^_4_ [39,46], and Zn^II^_4_ [39,41]; hexanuclear Co^II^_4_Co^III^_2_ [47] and Cu^II^_6_ [47], octanuclear Mn^II^_2_Mn^III^_6_ [48], and polynuclear Mn^III^ complexes [25,35]. Among these oligonuclear and polynuclear metal complexes, we have focused on the trinuclear metal systems important as the first step to polynucleation, especially the Co^II^Co^III^_2_ species because of the mixed-valent state. Another interesting point of cobalt complexes stems from the fact that cobalt(II) complexes have attracted much attention as good candidates for single-molecule magnets [49,50]. Although the trinuclear Co^II^Co^III^_2_ complex was prepared for 1,3-bis(salicylideneamino)-2-propanol, the reported magnetic susceptibility data were not analyzed [40], and there are no reports on such complexes with their substituted derivatives. This type of linear cobalt species was found in some related Co^II^Co^III^_2_ and Co^II^_3_ complexes and divided into five groups as shown in Figure 2: (a) Co^III^(octahedral)-Co^II^(tetrahedral)- Co^III^(octahedral) in [Co^II^{Co^III^(µ-L^1^)X_2_}_2_] (H_2_L^1^ = 1,3-bis(5-methyl-3-formylpyrazolylmethinimino)propane-2-ol, X = Cl, Br) [51]; (b) Co^III^(octahedral)-Co^II^(octahedral)-Co^III^(octahe dral) in [Co^II^{Co^III^(µ-L^2^)(µ-SO_3_)(C_3_H_7_OH)}_2_] (H_2_L^2^ = propane-1,3-dihylbis(α-methylsalicylideneiminate) [52], [Co^II^{Co^III^(µ-L^3^)(µ-CH_3_COO)(NCS)}_2_] (H_2_L^3^ = 1,6-bis(2- hydroxy phenyl)-2,5-diazahexa-1,5-diene) [53], [Co^II^{Co^III^(µ-L^4^)(µ-CH_3_COO)(NCS)}_2_] (H_2_L^4^ = 1,7-bis(2-hydroxyphenyl)-2,6-diazahepta-1,6-diene) [54], [Co^II^{Co^III^(µ-L^5^)(µ-CH_3_COO)(CH_3_COO)}_2_] (H_2_L^5^ = 1,6-bis(2-hydroxyphenyl)-2,5-diazahexa-1,5-diene or 2,7-bis(2-hydroxyphenyl)-2,6-diazaocta-2,6-diene) [55], and [Co^II^{Co^III^(µ-L^6^)(µ-CH_3_COO)(CH_3_COO)}_2_] (H_2_L^6^ = *N*,*N′*-bis(salicylidene)-meso-1,2-diphenylethylenediamine) [56]; (c) Co^II^(octahedral)-Co^II^(tetrahedral)-Co^II^(octahedral) in [Co^II^{Co^II^(µ-L^7^)_2_}_2_]X_2_ (HL^7^ = 2-[(3-aminopropyl)amino]ethanethiol; X = SCN, ClO_4_, NO_3_, Cl, Br, I) and [Co^II^{Co^II^(µ-L^8^)_2_}_2_]X_2_ (HL^8^ = 1-[(3-aminopropyl)amino]-2-methylpropane-2-thiol; X = NO_3_, ClO_4_, Cl, Br, I) [57,58]; (d) Co^II^(square-pyramidal)-Co^II^(octahedral)-Co^II^(square-pyramidal) in [Co^II^{Co^II^(µ-L^9^)(µ-CH_3_COO)}_2_] (H_2_L^9^ = 5,5′-dimethoxy-2,2′-[(ethylene)dioxybis(nitrilomethylidyne)]diphenol [59]; and (e) Co^II^(octahedral)-Co^II^(octahedral)-Co^II^(octahedral) in [Co^II^{Co^II^(µ-L^10^)(µ-CH_3_COO)(CH_3_COCH_3_)}_2_] (L^10^ = 4,4′-dichloro-2,2′-[(propane-1,3-dyldioxy)bis(nitrilomethylidyne)]diphenol) [60]. The Co^III^ oxidation state may come from the oxidation of Co^II^ by atmospheric oxygen during the reaction of Co^II^ salt and organic ligand [40,51,53,54,55]. The bridging µ-acetato ligand is favorable to form a trinuclear Co^III^-Co^II^-Co^III^ complex. From our synthesis experience, chloro-derivatives of Schiff-base ligands are promising for obtaining single crystals for X-ray crystallographic work [61]. To date, only two examples, (Et_4_N)[Mn^II^Mn^III^(clsalpr)_2_] [24] and [Mn^III^_2_(clsalpr)_2_(CH_3_OH)] [25], are known as metal complexes with the chloro-derivative of H_3_salpr, 1,3-bis(5-chlorosalicylideneamino)-2-propanol (H_3_clsalpr), which were structurally revealed by X-ray crystallography. In these complexes, the H_3_clsalpr ligand is fully deprotonated and works as a pentadentate ligand to two manganese centers to form dinuclear manganese complexes. In this study, we synthesized a new mixed-valent cobalt complex with a linear trinuclear Co^II^Co^III^_2_ core by the reaction of H_3_clsalpr (Figure 1) with cobalt(II) acetate tetrahydrate. The isolated complex was characterized by elemental analyses, IR and UV–vis spectroscopies, variable-temperature magnetic susceptibility measurements, and single-crystal X-ray structure analysis, elucidating the molecular structure of [Co_3_(Hclsalpr)_2_(CH_3_COO)_4_]. 

## 2. Results and Discussion

### 2.1. Synthesis of the Trinuclear Cobalt Complex 

The present complex was prepared by the reaction of 1,3-bis(5-chlorosalicylideneamino)-2-propanol (H_3_clsalpr) and cobalt(II) acetate tetrahydrate in acetonitrile at ambient temperature (Figure 3). As the cobalt salt, we selected cobalt(II) acetate tetrahydrate, aiming at the bridging property of acetate ions to form a trinuclear species. For a favorable condition for trinuclear formation, we reacted H_2_clsalpr with Co(CH_3_COO)_2_·4H_2_O in a 1:3 molar ratio under aerobic conditions, although we could isolate the same complex with a lower yield when the reaction was performed in a 1:1 or 1:2 molar ratio. The elemental analysis data of the obtained complex are in agreement with the trinuclear formulation of [Co_3_(Hclsalpr)_2_(CH_3_COO)_4_]. The oxidation of the two Co^2+^ ions to Co^3+^ ions may be accomplished by atmospheric oxygen acting as an oxidant, as usually observed for the synthesis of the related trinuclear Co^II^Co^III^_2_ complexes [40,51,53,54,55,56]. The synthetic method is similar to that of [Co_3_(Hsalpr)_2_(CH_3_COO)_4_] [40]. However, the reported method was slightly complicated with the further addition of an aqueous solution of NaN(CN)_2_ to the reaction solution.

### 2.2. Infrared Spectra of the Trinuclear Cobalt Complex

In the infrared spectrum of the complex, the C=N stretching band was observed at 1634 cm^−1^ due to the presence of the Schiff-base ligand. The lower energy shift compared with that of the free Schiff-base ligand (H_3_clsalpr: *ν*C=N at 1646 cm^−1^) suggests the coordination of the imino-nitrogen atom of the Schiff-base ligand in the cobalt complex. The complex shows two sets of antisymmetric stretching *ν*_as_(COO) and symmetric stretching *ν*_s_(COO) bands at 1590 and 1389 cm^−1^, respectively, with a ∆ value of 201 cm^−1^, and at 1562 and 1415 cm^−1^, respectively, with a ∆ value of 147 cm^−1^. The former and the latter may be ascribed to the typical IR spectral features of the monodentate and bridging acetate ligands, respectively [62,63]. These spectral features are similar to those of [Co_3_(Hsalpr)_2_(CH_3_COO)_4_] [40].

### 2.3. Electronic Spectra of the Trinuclear Cobalt Complex

The solid-state diffuse reflectance spectra exhibit a broad band with a lower-energy side shoulder at 354 nm, which may be ascribed to the CT transition band of the phenolate to metal as shown in Figure 4 [64]. The bands at 568 and 640 nm may be ascribed to d-d transitions (^1^*A*_1g_ → ^1^*B*_2g_, ^1^*A*_1g_ → ^1^*A*_2g_, ^1^*A*_1g_ → ^1^*E*_g_) of an octahedral Co^III^ with a low-spin state [64,65]. Furthermore, the spectra show a broad band at approximately 1260 nm, which can be ascribed to the d-d transition (^4^*T*_1g_ → ^4^*T*_2g_) due to an octahedral Co^II^ with a high-spin state [65]. The complex dissolves in THF. The solution spectra are similar to those of the solid-state spectra, showing a d-d absorption band (*ε* = 538 dm^3^ cm^−1^ mol^−1^) at 564 nm with a shoulder (*ε* = 320 dm^3^ cm^−1^ mol^−1^) at 632 nm, although absorption in the near-IR region could not be detected. Similar absorption spectra were reported for [Co_3_(Hsalpr)_2_(CH_3_COO)_4_] [40].

### 2.4. Crystal Structure of the Trinuclear Cobalt Complex 

Single crystals of the complex suitable for X-ray crystal structure analysis were grown by the slow evaporation of the THF solution of the complex. Crystallographic data are collected in Table 1. Selected bond distances and angles are given in Table 2. The complex crystallized in the monoclinic system. A perspective drawing of the structure is depicted in Figure 5. The molecule is a centrosymmetric trinuclear cobalt complex, where the Co1 atom is located at the crystallographical inversion center. The two Schiff-base ligands work as anionic tetradentate ligands Hclsalpr^2–^ to the terminal two cobalt atoms, Co2 and Co2^i^, where the superscript i denotes the equivalent position (1 − *x*, 1 − *y*, 1 − *z*), and the alcoholate hydrogen atom is not deprotonated, but two phenolate H atoms of each Schiff-base ligand are deprotonated. The Co1 atom is coordinated by two sets of two phenoxido-O atoms of Hclsalpr^2–^ ligands (O1, O3, O1^i^, O3^i^;) and µ-acetato-O atoms (O4 and O4^i^) to form an octahedral geometry with Co-O distances of 2.0493(16)–2.1318(15) Å. It should be noted that the axial bond lengths (2.1318(15) Å) are longer than the equatorial bond lengths (2.0493(16) and 2.0851(16) Å), showing an axial distortion around the Co1 atom. The Co2 atom is coordinated by two phenoxido-O atoms (O1 and O3) and two imino-N atoms (N1 and N2) of the tetradentate Schiff-base ligand in trans geometry [66] to occupy the equatorial site. The axial site is occupied by the O atoms of the µ-bridging acetate (O5) and monodentate acetate (O6). The Co-O and Co-N bond distances are in the range of 1.8943(15)–1.9263(16) Å, significantly shorter than those of the Co1 atom. The difference between the bond distances around the Co1 and Co2 (Co2^i^) atoms suggests that the Co1 atom is in a high-spin state of Co^2+^ ion and that the Co2 and Co2^i^ atoms are in a low-spin Co^3+^ ion state [67]. The bond valence sum calculation supports the mixed-valent Co^III^-Co^II^-Co^III^ state [68,69]. This is in agreement with the spectral feature in the diffuse reflectance spectra of the present complex. The alcoholate H atom of O2 is hydrogen bonded to the monodentate acetate-O atom O7 [O2-H…O7 2.659(2) Å]. In the crystal, there are four THF molecules in the asymmetric unit, and these molecules are oriented around the trinuclear molecule (Figure 6). The trinuclear structure is similar to that of the reported trinuclear cobalt complex [Co_3_(Hsalpr)(CH_3_COO)_4_], which lacks a center of symmetry, where the distortion around the Co^2+^ ion is more distorted compared with the present complex [40]. In these complexes, the two bridging acetate groups and four µ-phenoxido-O atoms of the Schiff-base ligands play an important role in connecting the two tetradendate Co(Hclsalpr)_2_ moieties. This motif was also found in the trinuclear zinc(II) complex [Zn_3_(Hsalpr)_2_(CH_3_COO)_2_] [38] and heterometallic trinuclear complexes [(CH_3_OH)_2_H^+^][NaMn_2_(Hsalpr)_2_(CH_3_COO)_2_] [25] and [Zn{Cu(salpd-µ-O,O’)(µ-CH_3_COO)}_2_] [H_2_salpd = propane-1,3-diylbis(salicylideneimine)] [10] as well as trinuclear cobalt complexes, [Co^II^{Co^III^(µ-L^5^)(µ-CH_3_COO)(CH_3_COO)}_2_] [55] and [Co^II^{Co^III^(µ-L^6^)(µ-CH_3_COO)(CH_3_COO)}_2_] [56]. 

### 2.5. Magnetic Properties of the Trinuclear Cobalt Complex

The present complex is expected to be paramagnetic because of the presence of the high-spin Co^2+^ ion at the central position of the trinuclear cobalt molecule, although the terminal two Co^3+^ ions are in a diamagnetic low-spin state. The magnetic susceptibility data for the complex are depicted in Figure 7 as the temperature variation of the *χ*_M_*T* product. The effective magnetic moment at 300 K is 5.73 µ_B_ per trinuclear molecule, which corresponds to the theoretical value of 5.20 µ_B_ for a magnetically isolated *S* = 3/2 spin with the contribution of orbital angular momentum (*L* = 3). The magnetic moment gradually decreases with decreasing temperature, reaching a value of 3.82 µ_B_ at 4.5 K. This magnetic behavior is similar to that of the related linear Co^III^-Co^II^-Co^III^ complex with 1,3-bis(salicylideneamino)-2-propanol [40]. The decrease in the magnetic moments may be ascribed to the axial distortion around the Co^2+^ ion, which was observed in the crystal structure. The axial splitting parameter ∆ was defined as the splitting of the local ^4^*T*_1g_ state of the octahedral Co^2+^ ion in the absence of spin–orbit coupling and introduced to the magnetic data analysis [70,71,72]. The magnetic data were simulated with the axial splitting parameter ∆, the spin-orbit coupling parameter *λ*, the orbital reduction factor *κ* for the Co^2+^ ion (H = Δ(L*_z_^2^* − 2/3) − (3/2)*κλ*L·S + *β*[–(3/2)*κ*L*_u_* + *g_e_*S*_u_*]·*H_u_* (*u* = *x*, *z*)), the temperature-independent paramagnetism *tip* for the Co centers, and the Weiss constant *θ* for intermolecular magnetic interactions by using the MagSaki(A)W1.0.11 program [72]. Magnetic susceptibility equations are shown below (Equations (1)–(6)), where $E_{n}^{\left(0\right)}$, $E_{u,n}^{\left(1\right)}$, and $E_{u,n}^{\left(2\right)}$ (*n* = ±1 − ±6, *u* = *x*, *z*) represent the zero-field energies, first-order Zeeman coefficients, and second-order Zeeman coefficients of the local ^4^*T*_1_ ground state for the octahedral Co^2+^ ion. From this, the anisotropic *g*-factors, *g*_z_ and *g*_x_, could be simulated using these parameters [72]. The simulation gave the following parameter values: Δ = –254 cm^−1^, *λ* = –58 cm^−1^, *κ* = 0.93, *tip* = 0.00436 cm^3^ mol^−1^, and *θ* = –0.469 K. A large value of the *tip* may be ascribed to the presence of three cobalt atoms in the molecule. The *g* values were simulated as *g*_z_ = 6.90 and *g*_x_ = 2.64. This result suggests that the magnetic behavior of the present complex can be interpreted by the axial distortion of the central Co^2+^ ion and thus proposed to be considerably anisotropic. If we apply the present magnetic analysis to the reported magnetic data of [Co_3_(Hsalpr)_2_(CH_3_COO)_4_] [40], we obtain the following parameter values: Δ = –950 cm^−1^, *λ* = –131 cm^−1^, *κ* = 0.93, *tip* = 0.00082 cm^3^ mol^−1^, *θ* = –0.67 K, *g*_z_ = 7.71, and *g*_x_ = 1.94, as shown in Figure 8. The magnetic analysis suggests that a considerable anisotropic character may also be found in [Co_3_(Hsalpr)_2_(CH_3_COO)_4_] and that the larger negative Δ and *λ* values may reflect a greater degree of axial distortion around the Co^2+^ ion in the crystal structure [40].
(1)$$\chi_{M}=\frac{\chi_{z}+2\chi_{x}}{3}$$
(2)$$\chi_{z}=N\frac{F_{1}}{F_{2}}+tip$$
(3)$$\chi_{x}=N\frac{F_{3}}{F_{2}}+tip$$
(4)$$F_{1}=\underset{n=\pm 1}{\sum }\left(\frac{E_{z,n}^{\left(1\right)}{}^{2}}{k\left(T-\theta \right)}-2E_{z,n}^{\left(2\right)}\right)\exp\left[\frac{-E_{n}^{\left(0\right)}}{kT}\right]+\underset{n\neq \pm 1}{\sum }\left(\frac{E_{z,n}^{\left(1\right)}{}^{2}}{kT}-2E_{z,n}^{\left(2\right)}\right)\exp\left[\frac{-E_{n}^{\left(0\right)}}{kT}\right]$$
(5)$$F_{2}=\underset{n}{\sum }\exp\left[\frac{-E_{n}^{\left(0\right)}}{kT}\right]$$
(6)$$F_{3}=\underset{n=\pm 1}{\sum }\left(\frac{E_{x,n}^{\left(1\right)}{}^{2}}{k\left(T-\theta \right)}-2E_{x,n}^{\left(2\right)}\right)\exp\left[\frac{-E_{n}^{\left(0\right)}}{kT}\right]+\underset{n\neq \pm 1}{\sum }\left(\frac{E_{x,n}^{\left(1\right)}{}^{2}}{kT}-2E_{x,n}^{\left(2\right)}\right)\exp\left[\frac{-E_{n}^{\left(0\right)}}{kT}\right]$$

### 2.6. Cyclic Voltammogram of the Trinuclear Cobalt Complex

The redox behavior of the complex was studied by cyclic voltammetry. The cyclic voltammogram (Figure 9) showed an irreversible reduction wave at approximately –1.2 V vs. Fc/Fc^+^, which may be assigned to the reduction of the terminal Co^3+^ ions in the reduction of the Co^III^-Co^II^-Co^III^ species. The corresponding oxidation wave can be observed at approximately –0.3 V vs. Fc/Fc^+^. No oxidation wave was observed until +1.0 V vs. Fc/Fc^+^ on the oxidation side. This result suggests that the trinuclear complex may not be maintained in the redox reaction, meaning that the stable form of the trinuclear species should be the Co^III^-Co^II^-Co^III^ mixed-valent state. A similar irreversible reduction wave was observed in [Co^II^{Co^III^(µ-L^1^)X_2_}_2_] [51] and [Co^III^_2_(nitrosalpr)_2_(CH_3_OH)] (H_3_nitrosalpr = 1,3-bis(5-nitrosalicylideneamino)-2-propanol) [29].

## 3. Materials and Methods

All reagents and metal salts were obtained from commercial sources and used without further purification.

The Schiff-base ligand H_3_clsalpr was prepared by the methods described in the literature [13,15,16]. An amount of 1,3-Diamino-2-propanol (2.177 g, 0.024 mol) and 5-chlorosalicylaldehyde (7.566 g, 0.048 mol) were dissolved in methanol (45 cm^3^). The solution was refluxed for 3 h and then left at room temperature overnight. The resulting yellow crystals were filtered off and recrystallized from methanol. Yield, 4.149 g (46%). IR (KBr, cm^−1^): ν(OH) 3130, ν(Ar-H) 3045, ν(C-H) 2891, ν(C=N) 1646.

Synthesis of [Co_3_(Hclsalpr)_2_(CH_3_COO)_4_]: To an acetonitrile solution (4 cm^3^) of H_3_clsalpr (36.7 mg, 0.1 mmol), Co(CH_3_COO)_2_·4H_2_O (74.7 mg, 0.3 mmol) and five drops of triethylamine were added. The solution was allowed to stand in a refrigerator, producing dark-brown crystals of **1** in 47% yield (26.6 mg) after several days. Anal. Found: C, 41.49; H, 3.90; N, 4.56%. Calcd for C_42_H_48_Cl_4_Co_3_N_4_O_18_ ([Co_3_(Hclsalpr)_2_(CH_3_COO)_4_]·4H_2_O): C, 41.50; H, 3.98; N, 4.61%. IR (KBr, cm^−1^): ν(OH) 3426, 3216, ν(Ar-H) 3020, ν(C-H) 2927, ν(C=N) 1634; ν_as_(COO) 1590, 1562, ν_s_(COO) 1415, 1389. Diffuse reflectance spectra: *λ*_max_ 354, 568, 640, 1264 nm.

Analytical data of C, H, and N were obtained on a Thermo Finnigan FLASH EA1112 series CHNO-S analyzer (Thermo Finnigan, Milan, Italy). IR spectra were obtained by KBr discs of samples on a JASCO MFT-2000 FT-IR spectrometer (JASCO, Tokyo, Japan). Powder reflectance spectra were obtained on a Shimadzu Model UV-3100 UV-vis-NIR spectrophotometer (Shimadzu, Kyoto, Japan). Magnetic susceptibility measurements were obtained on a Quantum Design SQUID susceptometer (MPMS-XL7, Quantum Design North America, San Diego, CA, USA) with a magnetic field of 0.5 T over a temperature range of 4.5–300 K. The magnetic susceptibility *χ*_M_ is the molar magnetic susceptibility per mole of [Co_3_(Hclsalpr)_2_(CH_3_COO)_4_] unit and was corrected for the diamagnetic contribution calculated from Pascal’s constants [73]. Cyclic voltammograms were measured in THF solutions containing tetra-*n*-butylammonium perchlorate (TBAP) on a BAS 100BW Electrochemical Workstation (Bioanalytical Systems, West Lafayette, IN, USA) with a glassy carbon electrode, a platinum wire counter electrode, and an Ag/Ag^+^ reference electrode. Ferrocene (Fc) was used as an internal standard. All the potentials are quoted relative to Fc^+^/Fc. 

X-ray crystallographic data were collected on a Bruker Smart APEX CCD diffractometer (Bruker, Billerica, MA, USA) using graphite monochromated Mo-Kα radiation. The structures were solved by intrinsic phasing methods and refined by full-matrix least-squares methods. The hydrogen atoms were included at their geometrical positions calculated geometrically. All of the calculations were carried out using the SHELXTL software package [74]. Crystallographic data have been deposited with Cambridge Crystallographic Data Centre: Deposit number CCDC-2175785. Copies of the data can be obtained free of charge via http://www.ccdc.cam.ac.uk/conts/retrieving.html (accessed on 30 May 2022) (or from the Cambridge Crystallographic Data Centre, 12, Union Road, Cambridge, CB2 1EZ, UK; Fax: +44 1223 336033; e-mail: deposit@ccdc.cam.ac.uk).

## 4. Conclusions

In this study, new trinuclear cobalt complex was synthesized by the reaction of 1,3-bis(5-chlorosalicylideneamino)-2-propanol (H_3_clsalpr) with cobalt(II) acetate tetrahydrate. The X-ray structure analysis revealed that a linear trinuclear Co^III^-Co^II^-Co^III^ complex was formed with two partially deprotonated Schiff-base ligands Hclsalpr^2–^, two bridging acetate ligands, and two monodentate acetate ligands. The electronic absorption spectra and cyclic voltammetry data suggest that the mixed-valent oxidation state is stable. The temperature dependence of the magnetic susceptibilities is in accordance with the magnetic property of the central Co^2+^ ion becoming considerably anisotropic due to the axial distortion of the coordination geometry. This anisotropic property could also be found in the related trinuclear complex [Co_3_(Hsalpr)_2_(CH_3_COO)_4_]. The anisotropic magnetic behavior of the mixed-valent Co^III^-Co^II^-Co^III^ complexes is interesting as a potential application for single-molecule magnets. Further study to pursue such magnetic relaxation properties is planned in our laboratories.

## Figures and Tables

**Figure 1 molecules-27-04211-f001:**
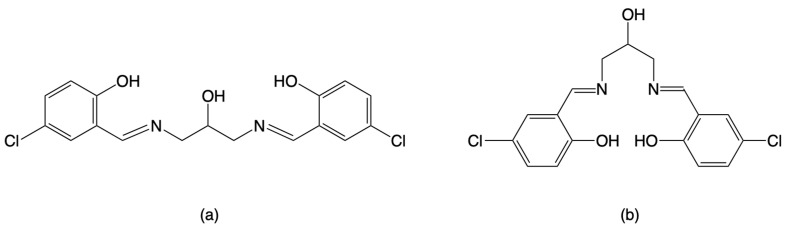
Schiff-base ligand 1,3-bis(5-chlorosalicylideneamino)-2-propanol as (**a**) pentadentate dinucleating ligand and (**b**) tetradentate mononucleating ligand.

**Figure 2 molecules-27-04211-f002:**
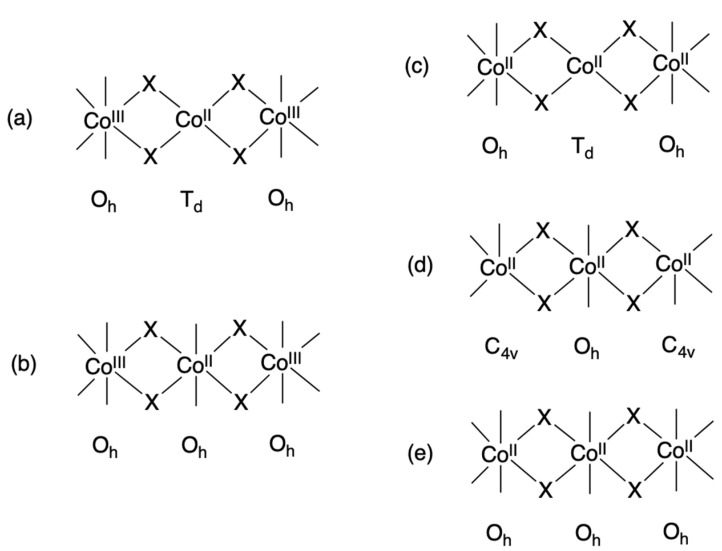
(**a**–**e**) Trinuclear cobalt complexes with a linear array of Co^III^-Co^II^-Co^III^ or Co^II^-Co^II^-Co^II^.

**Figure 3 molecules-27-04211-f003:**
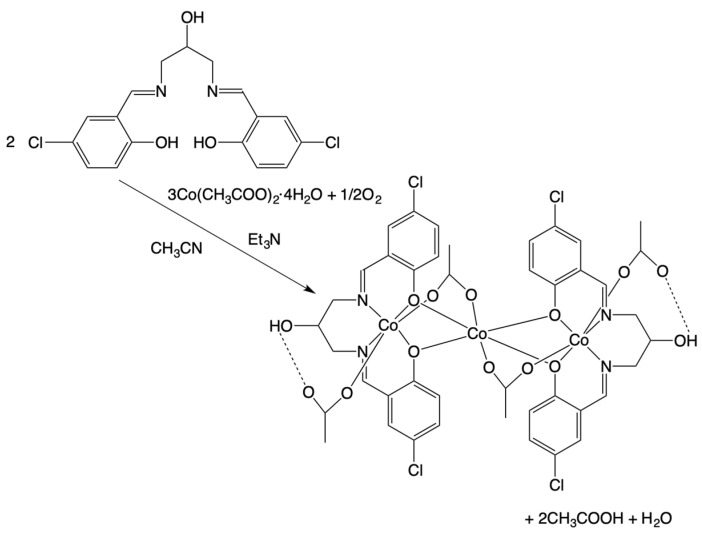
Synthetic scheme of the trinuclear cobalt complex [Co_3_(Hclsalpr)_2_(CH_3_COO)_4_].

**Figure 4 molecules-27-04211-f004:**
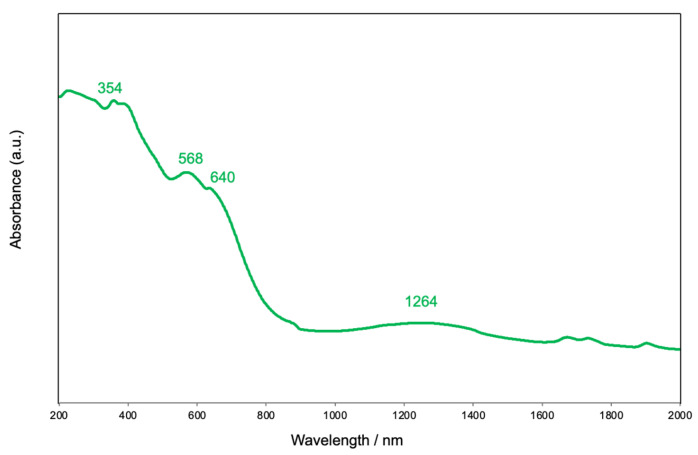
Diffuse reflectance spectra of [Co_3_(Hclsalpr)_2_(CH_3_COO)_4_] (green line).

**Figure 5 molecules-27-04211-f005:**
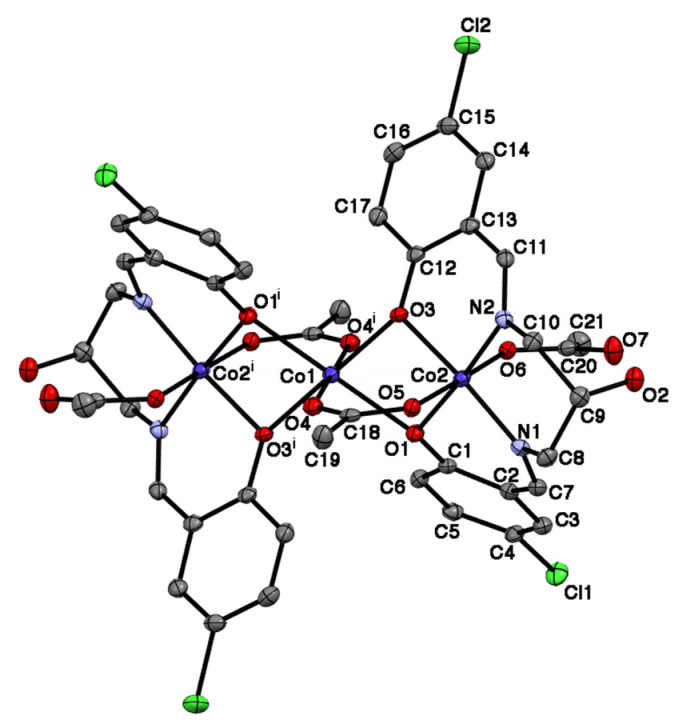
The ORTEP view of the molecular structure of [Co_3_(Hclsalpr)_2_(CH_3_COO)_4_] with thermal ellipsoids (50% probability level). The hydrogen atoms have been omitted for clarity.

**Figure 6 molecules-27-04211-f006:**
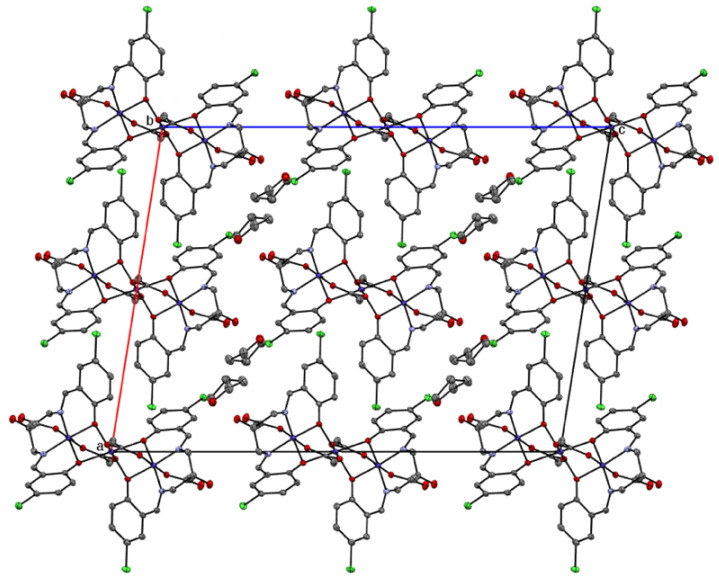
Packing diagram of [Co_3_(Hclsalpr)_2_(CH_3_COO)_4_]·8THF. The hydrogen atoms have been omitted for clarity.

**Figure 7 molecules-27-04211-f007:**
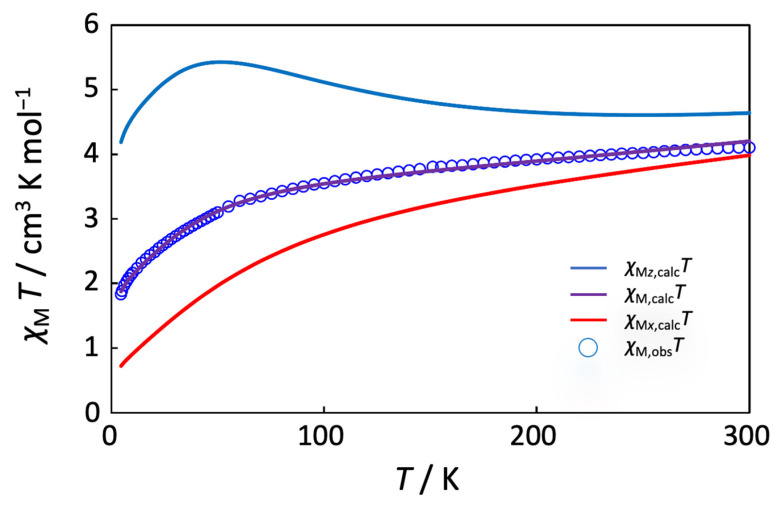
Variable temperature of *χ*_M_*T* for [Co_3_(Hclsalpr)_2_(CH_3_COO)_4_] (blue circle). The solid lines were calculated and drawn with the parameter values described in the text.

**Figure 8 molecules-27-04211-f008:**
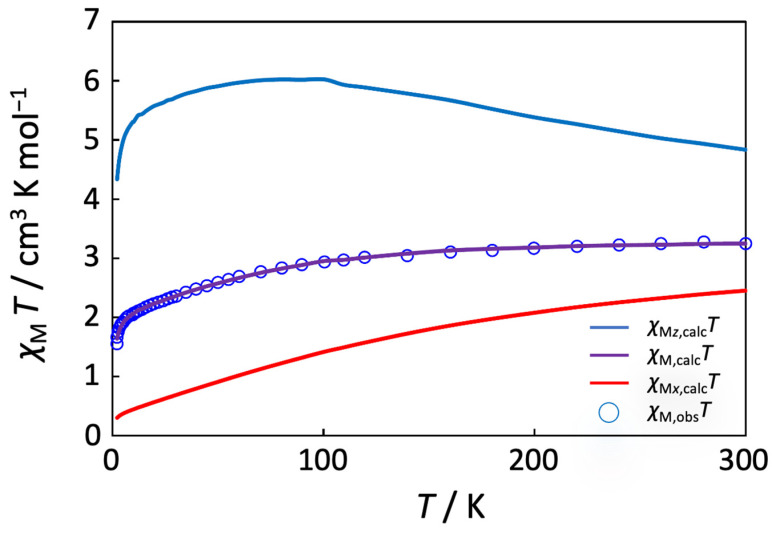
Variable temperature of *χ*_M_*T* for [Co_3_(Hsalpr)_2_(CH_3_COO)_4_] (blue circle) from the data in [40]. The solid lines were calculated and drawn with the parameter values described in the text.

**Figure 9 molecules-27-04211-f009:**
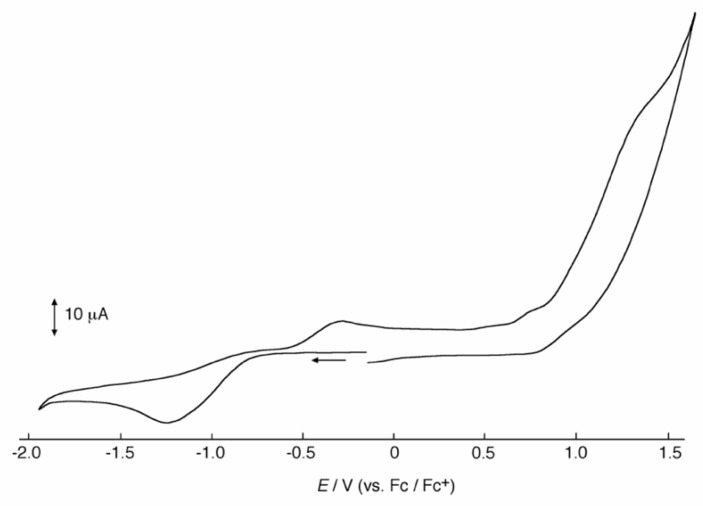
Cyclic voltammograms of [Co_3_(Hclsalpr)_2_(CH_3_COO)_4_] in THF ([complex] = 1 × 10^–3^ M; [TBAP] = 0.2 M; scan rate = 100 mV s^−1^).

**Table 1 molecules-27-04211-t001:** Crystallographic data and structure refinement.

Complex	[Co_3_(Hclsalpr)_2_(CH_3_COO)_4_]·8THF
Chemical formula	C_74_H_104_Cl_4_Co_3_N_4_O_22_
*FW*	1720.20
Temperature, *T* (K)	90
Crystal system	monoclinic
Space group	*C*2*/c*
*a* (Å)	20.244 (2)
*b* (Å)	14.0864 (16)
*c* (Å)	27.646 (3)
*β* (°)	98.9610 (10)
*V* (Å^3^)	7787.5 (14)
*Z*	4
*D*_calcd_ (g cm^−3^)	1.467
Crystal size (mm)	0.07 × 0.50 × 0.62
*μ* (mm^−1^)	0.845
*θ* range for data collection (°)	1.49–28.66
Reflections collected/unique	23,114/9103
[*R*1(*I* > 2σ(*I*)); *wR*2(all data)] ^(a)^	*R*_1_ = 0.0419
	*ωR*_2_ = 0.0953
GOF	1.003

^(a)^*R*1 = ∑||*F*_o_| − |*F*_c_||/∑|*F*_o_|; *ωR*2 = [∑*ω*(*F*_o_^2^ − *F*_c_^2^)^2^/∑(*F*_o_^2^)^2^]^1/2^.

**Table 2 molecules-27-04211-t002:** Selected bond distances (Å) and angles (°).

[Co_3_(Hclsalpr)_2_(CH_3_COO)_4_]·8THF	
Co1···Co2 3.0383(4)	Co2···Co2^i^ 6.0765(8)
Co1-O1 2.1318(15)	Co1-O3 2.0851(16)
Co1-O4 2.0493(16)	Co2-O1 1.9263(16)
Co2-O3 1.9220(15)	Co2-O5 1.9144(16)
Co2-O6 1.8943(15)	Co2-N1 1.9188(19)
Co2-N2 1.9173(19)	N1-C7 1.284(3)
N1-C8 1.476(3)	N2-C11 1.283(3)
N2-C10 1.473(3)	
O1-Co1-O1^i (a)^ 180.0	O1-Co1-O3 76.23(6)
O1-Co1-O3^i^ 103.77(6)	O1-Co1-O4 84.59(6)
O1-Co1-O4^i^ 95.41(6)	O3-Co1-O4 85.33(6)
O3-Co1-O4^i^ 94.67(6)	O1-Co2-O3 85.13(7)
O1-Co2-O5 91.70(7)	O1-Co2-O6 86.01(7)
O1-Co2-N1 89.97(7)	O1-Co2-N2 175.73(7)
O3-Co2-O5 91.70(7)	O3-Co2-O6 86.35(7)
O3-Co2-N1 175.08(7)	O3-Co2-N2 90.60(7)
O5-Co2-O6 176.96(7)	O5-Co2-N1 87.98(7)
O5-Co2-N2 88.60(7)	O6-Co2-N1 93.76(7)
O6-Co2-N2 93.74(7)	N1-Co2-N2 94.30(7)

^(a)^ i: the equivalent position (1 − *x*, 1 − *y*, 1 − *z*).

## Data Availability

Not applicable.

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
