# Peer review of "Mixed-Valent Trinuclear CoIII-CoII-CoIII Complex with 1,3-Bis(5-chlorosalicylideneamino)-2-propanol"

_molecules, 2022, doi:10.3390/molecules27134211_

Round 1
Reviewer 1 Report
The study reports on the synthesis and structural characterization of the mixed-valent cobalt complex with 1,3-bis(5-chlorosalicylideneamino)-2-propanol as a ligand. The manuscript can be accepted for publication after consideration of the following issues:
1) In the first sentence of the Abstract and in Conclusions, it was written that 1,3-bis(5-chlorosalicylideneamino)-3-propanol was used as a ligand, while through the rest of the manuscript, the name of the ligand is 1,3-bis(5-chlorosalicylideneamino)-2-propanol. This should be uniformed.
2) The second and third sentence in the abstract should be deleted.
3) The rationale for choice of cobalt as a metal ion for the synthesis of the complex should be included. Moreover, the authors should explain why they decided to synthesize the titled complex. What can be potential application of this complex?
4) A Scheme presenting the synthesis of the complex should be added.
5) What is solubility of the complex?
6) The solution UV-Vis spectra of the complex could be measured and compared with the solid-state reflectance spectra.
7) The chemical formula of the complex in Table 1 (C74H104Cl4Co3N4O22) is not in accordance with that in the Experimental section (C42H48Cl4Co3N4O18). Elemental analysis should be performed for the crystallized complex.
8) The part of the manuscript related to the cyclic voltammetry should be more detailed and can include the comparison of the redox behavior of the synthesized complex with that of the structurally similar complexes.
9) The Conclusion part should be more detailed.
Author Response
First, we would like to thank you for valuable suggestions for our manuscript.
1) The ligand name was written erroneously in the abstract and conclusion. We corrected this.
2) We agree with the suggestion and deleted the second and third descriptions.
3) According to the suggestion, the motivation of this study was included in the introduction.
4) According to the suggestion, we added the Scheme concerning to synthesis of the cobalt complex as Figure 3.
5) The complex dissolves in THF.
6) We added the description on the solution spectra of the complex in THF, which is similar to the solid-state spectra.
7) The crystals are efflorescent. So, we did not perform the elemental analysis for the crystals. The presence of the THF molecules are clear from the satisfied X-ray structure analysis.
8) We added some comment on the CV data.
9) We added more comment in the conclusion.
The corrected part was written in red color.
I hope that the revised manuscript will be OK.
Reviewer 2 Report
Although the complex reported has a similar molecular structure with an analogous complex, albeit with a slightly different ligand (ref. [40]), the scientific content of the present manuscript is interesting; thus, I propose its acceptance and publication in MOLECULES. I am sure that the paper will attract the interest of researchers working in the areas of : (a) the chemistry and properties of cobalt clusters (especially the mixed-valence ones), and (b) the coordination chemistry of polydentate Schiff bases. Also, I believe that the article will receive a respectable number of citations in the future. Salient features of this work – which support my proposal for acceptance – are: (1) The reported complex has a remarkable molecular structure, and (2) its magnetic properties are interesting and well interpreted. The ms is well written and the quality of figures is high. The references list covers the topic under study satisfactorily.
Based on the above mentioned I am glad because I can propose acceptance of this piece of research in MOLECULES. Minor revision points/comments/suggestions to be taken into account by the authors:
(1) Section 4 (Conclusions) : The perspectives of this work should be outlined.
(2) Part 2.4, line 143 : The neutral ligand should be abbreviated as H2salpd.
(3) Part 2.4 : I would welcome a more detailed description of the supramolecular structural characteristics of the compound; any π-π stacking interactions?
(4) Part 2.1 : The chemical equation illustrating the preparation of the complex should be written with the atmospheric oxygen as the oxidant.
(5) “Abstract” : The details about crystallographic data (crystal system, space group, …) should be deleted.
(6) Caption of Figure 2 : “Diffuse” instead of “diffused”.
(7) Part 2.5 : Did the authors check if the complex behaves as Single-Ion Magnet? A comment is needed at this point.
(8) Are there structurally characterized complexes with the trianionic clsalpr(-3) ligand? If yes, these should be briefly mentioned (preferably in a table) and discussed in terms of the dimensionality/nuclearity of the products and the coordination modes of the ligand.
(9) The magnetic properties of the present mixed-valence cluster should be compared with those of the analogous complex mentioned in ref. [40].
Author Response
First, we would like to thank you for valuable suggestions for our manuscript.
(1) We added some comments in the conclusion after your suggestions.
(2) According to the suggestion, we corrected the ligand abbreviation, and corrected the ligand name also.
(3) We looked into the crystal structure to see any supramolecular structure. Unfortunately we could not found such supramolecular feature.
(4) Thank you for pointing out this. According to the suggestion, we added a synthesis scheme as Figure 3.
(5) According to the suggestion, we deleted the crystallographic description in the abstract.
(6) We corrected this.
(7) We did not check the magnetic behavior concerning the single-ion magnet, because we could not have enough machine time to perform such a study. In this study, we confirmed the anisotropic magnetic properties of the present complex as well as the reported related complex. We added a comment for future work in the conclusion.
(8) As for clsalpr3– ligand, there are two reports on dinuclear manganese complexes with clsalpr3–. We added a comment on these complexes in the introduction.
(9) According to the suggestion, we analyzed the magnetic data of the related trinuclear complex, [Co3(Hclsalpr)2(CH3COO)4], by reading the literature data, and added a comment. During the revision process, we improved our magnetic analysis by applying the Weiss constant to the lowest energy levels, corrected equations (1)—(6), and thus corrected magnetic parameters.
The corrected part was written in red color.
I hope that the revised manuscript will be OK.
Reviewer 3 Report
Bis(salicylideneamino) derivatives are extensively exploited in the design of coordination compounds as Schiff base ligands due to their high potential to assemble diverse structure motifs and because the ease of introducing various functional properties. The authors are actively developing this field in the search new molecule-based magnetic materials and in the reviewed article molecules-1774185 describes the synthesis, structure, IR and UV-vis, electrochemical behavior and magnetic properties of trinuclear complex of mixed-valence cobalt(II/III) with 1,3-bis(5-chlorosalicylideneamino)-2-propanol ligand. They obtained high-quality experimental data using a classic set of well-proven methods that provides a thorough analysis of the valence state of Co central atoms in the complex. The described data should be of interest to specialists exploring the design of multifunctional coordination materials. However, the low level novelty and originality, the factual style of discussion without highlighting how research helps to solve any scientific problem and absence of some relevant references to previously published data allow to recommend this manuscript for publication only after major revision.
The following comments should be considered:
(I) Most importantly, in introduction, authors announced: “the trinuclear metal systems important as the first step to polynucleation, especially, the CoIICoIII2 species because of the mixed-valent state”. However, they did not discuss any of the factors favoring the formation of trinuclear molecules and the reason why is CoIIICoIICoIII oxidation state is ultimately stable.
Authors have to expand the set of references to data obtained for similar trinuclear cobalt complexes (e.g. 10.1016/j.poly.2006.08.035, 10.1002/ejic.200701025, 10.1016/j.ica.2008.06.023, 10.3390/cryst8010043, 10.3184/174751912X13366711594575, 10.1016/j.poly.2007.09.037). Comparison of the synthesis conditions, structural and UV-vis\IR features, and other physicochemical properties of trinuclear cobalt complexes can provide a deeper understanding of predictable design of these type of coordination oligomers with desirable valence states of Co2+/3+ ions.
Authors should add to the discussion how the synthesis conditions (the reagents ratio, the choice of the initial cobalt salt, temperature, pH, time etc.) affect the assembly of trinuclear and other species species. E.g. which compound oxidizes CoII to CoIII?
(IIa) In 2.6. Cyclic Voltammogram of the Trinuclear Cobalt Complex section, the discussion of the electrochemical behavior is too short. Why did not authors discuss the specifics of oxidation process?
(IIb) For all sections in Discussion, it is better to add a comparison with previously published data for related complexes.
Although I have been not able to find any typo/misprint, the manuscript, in my opinion, needs to be improved in some phrases and sentences (rewriting and shortening) and correcting some English grammar inelegancies.
line 32: change “transition metal elements” to “transition metal centers” or “transition metal ions”
lines 32,33: change “lanthanide and actinide elements” to “lanthanides and actinides”
lines 35-39: shorten sentence in order to avoid the repetition “dinuclea-“ 3 times.
lines 41-45: shorten sentence in order to avoid the repetition “dinuclea-“ 3 times.
line 173: change “calculated” to “simulated”
atoms and elements used are denoted by symbols from the periodic table (e.g., "O”, “H” instead of “oxygen”, “hydrogen” etc).
it is better to use only abbreviations for solvents (e.g. THF, as at the line 214, instead of “tetrahydrofuran”, as at lines 99, 123).
and others.
Author Response
First, we would like to thank you for valuable suggestions for our manuscript.
(I) Thank you for the valuable information on the related studies on trinuclear cobalt complexes. We added some comments introducing these studies. We also added a comment on the synthesis conditions.
(IIa) We added a comment on the cyclic voltammogram according to the suggestion.
(IIb) According to the suggestion, we added a description of comparison with the related data.
Thank you for pointing out English phrases and sentences. We corrected these and checked other part also.
According to the suggestion on element symbols and THF, we corrected these.
The corrected part was written in red color.
I hope that the revised manuscript will be OK.
Round 2
Reviewer 1 Report
The manuscript has been improved and it is acceptable in its present form.
Author Response
Thank you very much for your kind understanding for our revision.
Reviewer 3 Report
The text of the
manuscript became more suitable for publication after the author's amendments.
The level of novelty and originality has increased after the addition of some
explanatory comments in the sections "Introduction" and
"Conclusion" and links to previously published data. However, the corrected version contains typos and misprints, and some of
the text-inserts are too cumbersome. I recommend this manuscript for
publication with minor revisions, mainly related to the technical editing of
the text. (Lines 67-89: shorten the sentence to avoid the repetition of
"with a linear array." It is enough to indicate the general
"linear array" feature at the beginning of the sentence, and from my
point of view, the Co3 complexes can be combined into five groups, as shown in
Fig. 2. At discussion of ions it is better use the charge symbols instead of valences
– “Co3+ ions” instead of “cobalt(III) ions”.)

Author Response
We sould like to thank you for valuable suggestions for our revised manuscript.
According to the suggestions, we shortened the part of the introduction and corrected careless mistakes also.
The corrected part was written in brown color.
I hope that the revised manuscript will be OK.